# Performance of Coronary Angiography in the Detection of Coronary Artery Disease in Patients with Systolic Left Ventricular Dysfunction and No Prior Ischemic Heart Disease

**DOI:** 10.3390/jcm11041097

**Published:** 2022-02-18

**Authors:** Óscar M. Peiró, Maria Ferrero, Alba Romeu, Anna Carrasquer, Gil Bonet, Mohsen Mohandes, Alberto Pernigotti, Alfredo Bardají

**Affiliations:** 1Department of Cardiology, Joan XXIII University Hospital, 43005 Tarragona, Spain; opi220290@gmail.com (Ó.M.P.); mferrero.hj23.ics@gencat.cat (M.F.); aromeu.hj23.ics@gencat.cat (A.R.); carrasquer1987@gmail.com (A.C.); gil.bonet.p@gmail.com (G.B.); mohandesmohsen@hotmail.com (M.M.); a.pernigotti@gmail.com (A.P.); 2Pere Virgili Health Research Institute (IISPV), 43005 Tarragona, Spain; 3Department of Medicine and Surgery, Rovira i Virgili University, 43003 Tarragona, Spain

**Keywords:** ventricular dysfunction, coronariography, diagnosis, prognostication

## Abstract

The diagnosis of ischemic cardiomyopathy is not well established. Our objective is to determine predictive variables of coronary disease in unselected patients with ventricular dysfunction. This study is a retrospective cohort study of consecutive patients with left ventricular dysfunction and no known history of ischemic heart disease. We analyse the demographic variables, clinical data, electrocardiogram, and echocardiogram that are associated with the presence of coronary stenosis on coronary angiography. A total of 536 patients with left ventricular dysfunction were studied, with 135 (25.2%) of them having significant coronary lesions. In the multivariate logistic regression analysis, age ≤ 50 years, female gender, and the presence of atrial fibrillation on the electrocardiogram (ECG) were predictors of the absence of coronary lesions. Diabetes, hypercholesterolemia, the existence of Q waves in the ECG, and segmental alterations in contractility in the echocardiogram were predictors of coronary heart disease (C-Statistics 0.771, 95% CI 0.727 to 0.814). The information obtained from the clinical history, the ECG, and the echocardiogram of patients with ventricular dysfunction allows us to select subjects in whom coronary angiography has shown poor performance in diagnosing coronary disease.

## 1. Introduction

In the initial evaluation of patients with heart failure and systolic ventricular dysfunction, it is essential to determine the aetiological cause [1,2]. Although the treatment of heart failure is comparable among different aetiological entities, detecting that ischemia is the aetiological cause of ventricular dysfunction can have critical clinical repercussions [3]. The ischemic aetiology of ventricular dysfunction can be suspected by examining the data drawn from clinical records [4], cardiac imaging techniques [5,6], or coronary angiographic studies showing stenotic coronary lesions.

Coronary angiography is a frequently performed diagnostic technique in patients with systolic ventricular dysfunction and has been considered the technique of choice for confirming the presence of coronary disease. In current clinical practice guidelines, coronary angiography has an indication of IIb and a level of evidence of B. Still, this indication is not supported by specific clinical studies that have evaluated its diagnostic performance [1]. Despite these recommendations, there is significant variability in the practice of coronary angiography in patients with systolic ventricular dysfunction or heart failure [7,8,9]. This is mainly due to the lack of reliable data demonstrating this technique’s clear benefit in this population.

We hypothesize that coronary angiography in unselected patients with left ventricular systolic dysfunction should be performed only in patients with higher risk factors and should be avoided in those without risk factors. Therefore, the purpose of our study is to analyse the diagnostic performance of coronary angiography and to determine the predictive clinical variables of coronary disease in these patients.

## 2. Materials and Methods

This retrospective cohort study included all adult patients referred for diagnostic coronary angiography between 1 June 2008 and 31 December 2019, with a diagnosis of left systolic ventricular dysfunction (ejection fraction (EF)) less than 50%) detected by echocardiography. These patients were identified using the database of the Catheterization Laboratory. Patients with a known prior diagnosis of ischemic heart disease were excluded.

We collected demographic data, main cardiovascular risk factors (smoking, hypertension, diabetes, hypercholesterolemia, and obesity), diagnosis of previous heart failure, electrocardiographic (ECG) data (presence of atrial fibrillation, pathological Q waves, or interventricular conduction disorder), renal function (estimated glomerular filtration rate assessed by Chronic Kidney Disease Epidemiology Collaboration (CKD-EPI) formula), and basic data from the echocardiogram (EF and segmental changes in contractility).

All coronary angiographic studies were retrospectively analysed by an expert interventional cardiologist. In each study, we evaluated the presence of coronary artery disease of the left main coronary artery or disease of one, two, or three vessels based on the detection of coronary lesions of 50% or higher evaluated quantitatively.

The patients were assigned to two groups according to the presence or absence of significant stenosis on coronary angiography. In addition, patients with coronary stenosis were categorized into two groups: coronary artery disease of one vessel, and coronary artery disease of 2 or more vessels.

The sample size was calculated by estimating the risk factors for coronary stenosis in the first 50 patients included. Thus, the estimated sample size was 516 patients. Categorical variables were expressed as numbers and percentages, whereas continuous variables were expressed as the median and interquartile range. Categorical data were compared with the chi-squared test or Fisher’s exact test, while numerical data were analysed with the Mann–Whitney U test. We performed univariate and multivariable logistic regressions to determine which variables were associated with significant coronary stenosis. In the multivariable analysis, clinically relevant and significant variables in the univariable analysis were included. Therefore, multivariable logistic regression analysis was adjusted by age ≤50 years, female sex, past or current smoker, hypertension, diabetes mellitus, hypercholesterolemia, atrial fibrillation/flutter, presence of q- wave, estimated glomerular filtration rate (eGFR) ≤30 mL/min per 1.73 m^2^, and segmentary alterations. Multivariable logistic regression analysis was performed with the backward stepwise procedure.

According to the odds ratio (OR) attained by the clinical variables emerging at the multivariate analysis, we constructed a score system to predict significant coronary stenosis in patients with heart failure (SCS-HF score). Thus, the odds ratio of each variable was rounded to the nearest 0.5 multiple and then converted into points. Therefore, patients were classified into 3 risk categories: low risk: ≤−2 points; intermediate risk: >−2 points to ≤6 points; and high risk: >6 points. The area under the receiver operating characteristic (ROC) curve was assessed to determine the discriminative power and the goodness of fit was calculated using the Hosmer–Lemeshow test. The exact cut-off point for each category was determined on the basis of these criteria after careful analysis of the ROC curve. The best cut-off point for maximum efficiency was >6 points (high-risk patients).

Differences were considered statistically significant at *p* < 0.05. The STATA 14.2 software (StataCorp, College Station, TX) was used for statistical analysis.

## 3. Result

From a total number of 20,466 patients subjected to coronary angiography, we selected 536 patients for the study. Significant coronary lesions were detected in approximately a quarter of patients (*n* = 135, 25.2%; Figure 1).

### 3.1. Clinical Differences between Patients with and without Coronary Lesions

As summarized in Table 1, patients with coronary lesions more often had an age >50 years, were of the male sex, and exhibited cardiovascular risk factors than patients free of coronary lesions. Likewise, the incidence of pathological Q waves in the ECG was also higher in the group with coronary lesions, but atrial fibrillation was more present in the group free of coronary lesions. Left ventricular (LV) ejection fraction was depressed in both groups, but LV segmentary abnormalities were more frequent in the group with coronary stenosis. Renal function, assessed by glomerular filtration rate, was significantly worse in patients with coronary lesions.

### 3.2. Characteristics of Patients with Multi-Vessel Lesion

Among patients with coronary lesions, 54 (40%) corresponded to one-vessel disease, 39 (28.9%) corresponded to two-vessel disease, and 42 (31.1%) corresponded to three-vessel disease. Left main coronary artery disease was present in 13 patients (9.6%), and among them, 3 patients had associated one-vessel disease, 4 had two-vessel disease, and another 6 had three-vessel disease. The comparison between patients with one vessel, and patients with disease in two or more vessels did not show differences concerning age, sex, cardiovascular history, history of heart failure, electrocardiographic pattern, renal function, or echocardiogram abnormalities (Table 2).

### 3.3. Predictors of Coronary Heart Disease

Table 3 summarizes the clinical variables associated with the presence of coronary disease in the univariate and multivariate logistic regression analysis. Diabetes, hypercholesterolemia, presence of abnormal Q waves in the ECG, and segmentary alterations in the echocardiogram were predictors of coronary disease, whereas age ≤50 years and female gender were found in patients with an absence of coronary lesions.

The assignment of a score to the predictive or protective variables of coronary heart disease, based on the OR of the multivariate analysis, allowed us to establish a grading system for the observed probability (SCS-HF scoring; Figure 2). Thus, with a score ≤−2 (which was detected in 16.6% of the patients), the probability of coronary heart disease was anecdotal. In comparison, in patients with a score higher than 6 (9.7% of the population), the presence of coronary disease was very prevalent. Scores between −2 and 6 points showed an intermediate probability of coronary disease, between 12.9% and 36.5% (Figure 2 and Appendix A). The differences in the predictive variables of coronary heart disease in the three risk groups for coronary heart disease are shown in Table 4. Thus, age ≤ 50 years, atrial fibrillation, and female sex were the main variables observed in the group with a low probability of coronary disease. On the other hand, hypercholesterolemia, diabetes, segmental alterations in contractility in the echocardiogram and the presence of Q waves were the most prevalent variables in patients with a high probability of coronary disease. This predictive model provided an adequate discriminatory power (C-Statistics 0.771, 95% CI: 0.727–0.814). Assuming a prevalence of significant coronary stenosis similar to our study (25%), an SCS-HF score of >6 points represents a negative predictive value of 80.0% and a positive predictive value of 72.4%.

## 4. Discussion

Our study shows that the diagnostic yield of coronary angiography when detecting significant coronary artery disease in patients with systolic LV dysfunction and no prior history of ischemic heart disease is low if the procedure is performed in a nonselective fashion. Moreover, we identified a group of clinical variables that would allow for the identification of patients with a higher probability of presenting significant coronary disease during an angiographic study.

According to the results of our study, patients with a high score are at high risk of significant coronary stenosis and should be evaluated by coronary angiography while patients with a low score should avoid coronary angiography due to their low risk of coronary stenosis. However, whether coronary angiography should be used for patients with an intermediate score should be determined on a case-by-case basis, and non-invasive testing may be required before coronary angiography.

There is no single agreed-upon classification system for establishing the causes of ventricular dysfunction, and there is a large overlap between the potential categories. Some patients may have several different pathologies, cardiovascular and non-cardiovascular, which can potentially cause ventricular dysfunction. In addition, many patients with ischemic ventricular dysfunction have a history of myocardial infarction or revascularization. Even though some authors have pointed out that coronary angiography is the gold standard technique in patients with ventricular dysfunction [10], it is important that the demonstration of “significant” coronary artery disease does not necessarily imply causation [11]. In the Deschroche series, a total of 355 patients with an ejection fraction of less than 45% of unknown origin underwent cardiac resonance and coronary angiography. The presence of coronary lesions on the coronary angiograph was concentrated in patients with scars on cardiac resonance, in such a way that this last technique demonstrated a negative predictive value of 98% and a positive predictive value of 58% for the detection of coronary lesions in coronary angiography [5]. Thus, coronary angiography alone is not sufficient for diagnosing ischemic cardiomyopathy. In The Danish Study to Assess the Efficacy of ICDs in Patients with Non-ischemic Systolic Heart Failure on Mortality (DANISH), for example, non-ischemic cardiomyopathy was generally diagnosed by coronary angiography, by coronary normal computed tomographic (CT) angiogram, or by the absence of alterations in an isotopic perfusion study [12]. However, non-ischemic cardiomyopathy was also considered in patients with one- or two-vessel coronary disease if the extent of the coronary disease did not reasonably explain the ventricular dysfunction. We detected that single-vessel coronary disease was present in 40% of our patients with significant coronary stenosis. However, without other medical history or imaging techniques, this information does not classify these patients as ischemic cardiomyopathic.

In our study, the combination of clinical variables, the ECG, the echocardiograph, and laboratory tests allowed for a reasonable estimate of the probability of coronary disease in patients with ventricular dysfunction and, more importantly, could suggest to a clinician that the use of this invasive diagnostic technique should be avoided in a notorious proportion of patients. Other studies, based on a very small series of patients, have tried to identify predictors of coronary disease in left systolic ventricular dysfunction [13,14,15]. Thus, Smilowitz, analysing a series of patients much more limited than ours, observed that age, hypertension, diabetes, smoking, the presence of Q waves in the ECG, and abnormalities in the T wave or the ST segment were variables that allowed for estimating the presence of coronary heart disease in a predictive model [4]. Interestingly, similar to our data, the presence of a left bundle branch block was more prevalent in patients with non-ischemic cardiomyopathy (15%) than in patients with ischemic cardiomyopathy (2%, significant difference *p* = 0.03).

Our study supports the current recommendations of the European Society of Cardiology’s clinical practice guidelines on heart failure [1]: the indication for coronary angiography, with indication level IIb, should be selected for patients with heart failure with reduced ejection fraction, intermediate or high pre-test probability of coronary artery disease, and the presence of ischemia in non-invasive ischemia tests. However, the level of evidence for this recommendation is Level B as there are very few studies to date that have evaluated the diagnostic performance of this recommendation. This recommendation in the current European guidelines comes with a citation that corresponds to the Stich study [16]. In this randomized study (occurring between July 2002 and May 2007), a total of 1212 patients with an ejection fraction of 35% or less and coronary artery disease were subjected to bypass surgery, medical therapy alone (602 patients), or medical therapy plus revascularization (610 patients). The primary endpoint was the rate of death from any cause. The primary outcome occurred in 244 patients (41%) in the medical therapy group and 218 (36%) in the surgical group (risk index with surgeries 0.86; 95% confidence interval (CI), 0.72 to 1.04; *p* = 0.12). However, despite the primary endpoint not being demonstrated, patients assigned to surgery, compared with those assigned to medical therapy alone, had lower rates of cardiovascular death, or hospitalization (all secondary endpoints). In an extension of the Stich study to 9.8 years [17], revascularization surgery was associated with a reduction in mortality, total cardiovascular mortality, and readmissions for heart failure, compared to patients treated with drugs. Thus, in selected patients with ventricular dysfunction and coronary anatomy susceptible to surgical revascularization and with a long-term perspective, coronary surgery could be indicated to reduce total mortality and cardiovascular events. Despite the conclusions of the Stich study [16], the performance of coronary angiography in detecting coronary stenosis in patients with heart failure was not evaluated and no score was presented to predict the presence of coronary stenosis.

There are also two crucial reasons to investigate a possible ischemic cause in patients with ventricular dysfunction. First, the demonstration of coronary disease by any invasive or non-invasive technique is a reason for the installation of vigorous cardiovascular prevention measures because this group of patients is considered at high risk. Thus, antiplatelet therapy and high-intensity lipid-lowering therapy are fully justified to achieve a low-density lipoprotein (LDL) cholesterol level lower than 55 mg/dl and a decrease of more than 50% concerning base LDL cholesterol [18]. On the other hand, the demonstration of an ischemic origin of cardiomyopathy in patients with EF less than 35% and a functional class equal to or greater than II is a Class IA indication for the implantable cardioverter defibrillator (ICD) in current European clinical practice guidelines, unlike patients with non-ischemic cardiomyopathy who have an indication of IIaB [1].

Our study has several limitations. First, it was a series carried out in a single centre, so the conclusions cannot be generalized to other populations. Second, the number of variables analysed was limited and other unrecorded variables could modify the observed results. Third, the score for predicting coronary artery disease does not have internal or external validation. In this sense, the limited size of the sample does not allow us to conduct an internal validation. Finally, the presence of coronary lesions, especially in the case of a single affected vessel, does not necessarily imply the diagnosis of cardiomyopathy of ischemic origin.

## 5. Conclusions

Our study demonstrates that data from clinical history, ECG, and echocardiogram can be used to determine the risk stratification of coronary disease in patients with left ventricular systolic dysfunction. In this way, our risk score allows us to identify patients who should avoid coronary angiography and those who could benefit from it. In this sense, our results fully support, with objective data, indication IIb for coronary angiography in the current clinical practice guidelines of the European Society of Cardiology.

## Figures and Tables

**Figure 1 jcm-11-01097-f001:**
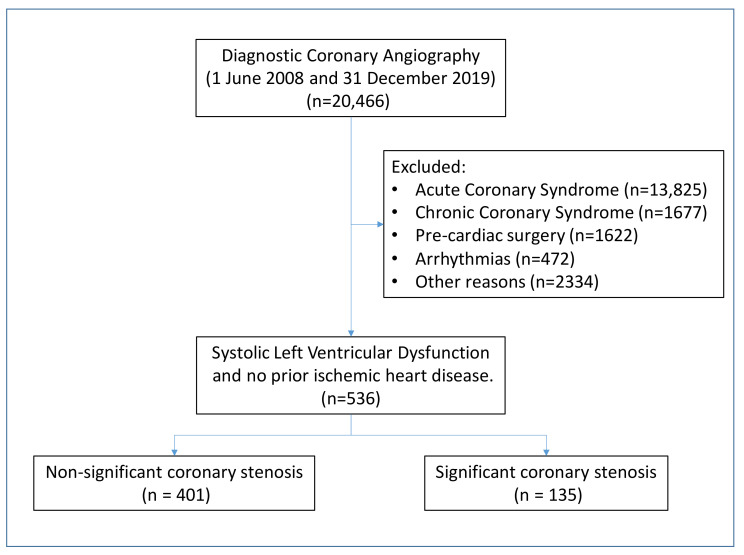
Flow diagram of patients. The distribution of patients in the two groups of the study is depicted.

**Figure 2 jcm-11-01097-f002:**
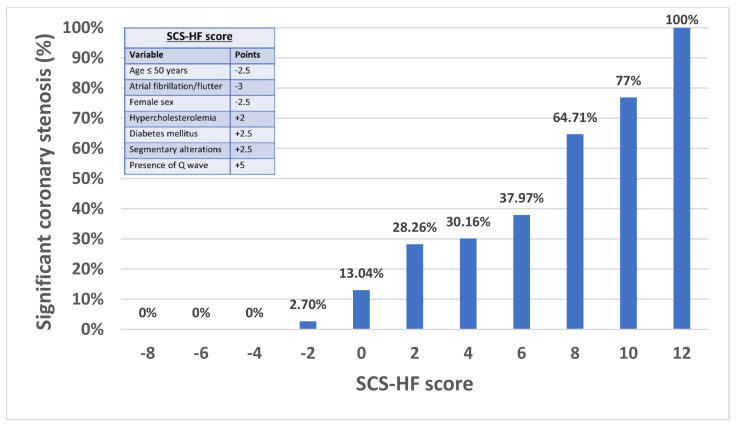
Significant coronary artery stenosis (%) across a score system to predict significant coronary stenosis in patients with heart failure (SCS-HF).

**Table 1 jcm-11-01097-t001:** Baseline characteristics by non-significant and significant coronary stenosis.

Variable	Overall (N = 536)	Non-Significant Coronary Stenosis (N = 401)	Significant Coronary Stenosis (N = 135)	*p* Value
Demographics				
Age, years	64.4 (56.5–71.8)	64.3 (55.5–71.9)	64.7 (58.0–71.5)	0.189
Age ≤ 50 years	63 (11.8)	57 (14.2)	6 (4.4)	0.002
Female sex	114 (21.3)	97 (24.2)	17 (12.6)	0.004
Cardiovascular risk factors				
Past or current smoker	335 (62.5)	235 (58.6)	100 (74.1)	0.001
Hypertension	377 (70.5)	273 (68.1)	104 (77.6)	0.036
Diabetes mellitus	207 (38.6)	127 (31.7)	80 (59.3)	<0.001
Hypercholesterolemia	244 (45.5)	161 (40.2)	83 (61.5)	<0.001
Obesity (BMI ≥ 30 kg/m^2^)	159 (33.1)	120 (33.2)	39 (33.1)	0.984
Medical history				
Heart failure	106 (19.9)	73 (18.3)	33 (24.6)	0.109
ECG				
Atrial fibrillation/flutter	110 (20.5)	95 (23.7)	15 (11.1)	0.002
Left bundle branch block	171 (32.4)	135 (34.1)	36 (27.3)	0.147
Right bundle branch block	36 (6.8)	25 (6.3)	11 (8.3)	0.425
Presence of Q wave	46 (8.7)	20 (5.0)	26 (19.7)	<0.001
Laboratory findings				
eGFR (mL/min per 1.73 m^2^)	79.4 (59.7–93.9)	81.1 (60.1–95.7)	76.8 (56.5–87.6)	0.011
Echocardiography				
Left ventricular ejection fraction	31 (25–37)	31 (25–37)	30 (25–37)	0.547
Segmentary alterations	136 (26.5)	78 (20.3)	58 (45.0)	<0.001

Data represent the number (percentage) or median (interquartile range). BMI indicates body mass index. eGFR indicates estimated glomerular filtration rate.

**Table 2 jcm-11-01097-t002:** Baseline characteristics by single or multi-vessel disease.

Variable	Overall (N = 135)	1-Vessel Disease (N = 54)	2- or 3-Vessel Disease (N = 81)	*p* Value
Demographics				
Age, years	64.7 (58.0–71.5)	64.6 (56.7–70.4)	64.9 (60.7–72.6)	0.305
Age ≤ 50 years	6 (4.4)	3 (5.6)	3 (3.7)	0.683
Female sex	17 (12.6)	10 (18.5)	7 (8.6)	0.090
Cardiovascular risk factors				
Past or current smoker	100 (74.1)	36 (66.7)	64 (79.0)	0.109
Hypertension	104 (77.6)	43 (79.6)	61 (76.3)	0.645
Diabetes mellitus	80 (59.3)	32 (59.3)	48 (59.3)	1.000
Hypercholesterolemia	83 (61.5)	38 (70.4)	45 (55.6)	0.083
Obesity (BMI ≥30 kg/m^2^)	39 (33.1)	15 (31.3)	24 (34.3)	0.731
Medical history				
Heart failure	33 (24.6)	13 (24.1)	20 (25.0)	0.903
ECG				
Atrial fibrillation/flutter	15 (11.1)	4 (7.4)	11 (13.6)	0.264
Left bundle branch block	36 (27.3)	16 (30.2)	20 (25.3)	0.538
Right bundle branch block	11 (8.3)	5 (9.4)	6 (7.6)	0.755
Presence of Q wave	26 (19.7)	9 (17.0)	17 (21.5)	0.520
Laboratory findings				
eGFR (mL/min per 1.73 m^2^)	76.8 (56.5–87.6)	74.1 (52.9–84.1)	77.6 (59.9–90.2)	0.329
Echocardiography				
Left ventricular ejection fraction	30 (25–37)	30 (25–38)	30 (27–36)	0.652
Segmentary alterations	58 (45.0)	20 (37.7)	38 (50.0)	0.168
Angiography				
Significant coronary stenosis in left main coronary artery	13 (9.6)	3 (5.6)	10 (12.4)	0.190

Data represent the number (percentage) or median (interquartile range). BMI indicates body mass index. eGFR indicates estimated glomerular filtration rate.

**Table 3 jcm-11-01097-t003:** Odds ratios associated with the presence of significant coronary stenosis in univariate and multivariate logistic analysis.

	Univariate Analysis		Multivariate Analysis	
Variables	OR (95% CI)	*p*-Value	OR (95% CI)	*p*-Value
Age ≤ 50 years	0.28 (0.12–0.67)	0.004	0.38 (0.15–0.98)	0.045
Female sex	0.45 (0.26–0.79)	0.005	0.40 (0.22–0.75)	0.004
Past or current smoker	2.02 (1.31–3.11)	0.001	-	-
Hypertension	1.63 (1.03–2.57)	0.037	-	-
Diabetes mellitus	3.14 (2.10–4.69)	<0.001	2.33 (1.46–3.72)	<0.001
Hypercholesterolemia	2.38 (1.59–3.55)	<0.001	2.22 (1.38–3.56)	0.001
Atrial fibrillation/flutter	0.40 (0.22–0.72)	0.002	0.31 (0.16–0.59)	<0.001
Presence of Q wave	4.62 (2.48–8.61)	<0.001	4.85 (2.30–10.22)	<0.001
eGFR ≤ 30 mL/min per 1.73 m^2^	3.09 (1.14–8.41)	0.027	-	-
Segmentary alterations	3.22 (2.10–4.93)	<0.001	2.47 (1.52–4.00)	<0.001

OR: odds ratio; CI: confidence interval; eGFR: estimated glomerular filtration rate.

**Table 4 jcm-11-01097-t004:** Numbers and percentages of patients in low, intermediate, and high risk according to the variables included in a score system to predict significant coronary stenosis in patients with heart failure (SCS-HF).

Variable	Overall (N = 536)	Low Risk (N = 89)	Intermediate Risk (N = 395)	High Risk (N = 52)	*p* Value
Age ≤ 50 years	63 (11.8)	37 (41.6)	26 (6.6)	0 (0.0)	<0.001
Atrial fibrillation/flutter	110 (20.5)	38 (42.7)	69 (17.5)	3 (5.8)	<0.001
Female sex	114 (21.3)	41 (46.1)	70 (17.7)	3 (5.8)	<0.001
Hypercholesterolemia	244 (45.5)	6 (6.7)	194 (49.1)	44 (84.6)	<0.001
Diabetes mellitus	207 (38.6)	2 (2.3)	160 (40.5)	45 (86.5)	<0.001
Segmentary alterations	136 (25.4)	3 (3.4)	89 (22.5)	44 (84.6)	<0.001
Presence of Q wave	46 (8.6)	0 (0.0)	20 (5.1)	26 (50.0)	<0.001

Low risk: ≤−2 points; intermediate risk: >−2 points to ≤6 points; high risk: >6 points.

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
