# Peer review of "Performance of Coronary Angiography in the Detection of Coronary Artery Disease in Patients with Systolic Left Ventricular Dysfunction and No Prior Ischemic Heart Disease"

_jcm, 2022, doi:10.3390/jcm11041097_

Round 1

Reviewer 1 Report

Dear authors,
First of all, I would like to thank you for submitting your work.
The proposed manuscript represents very good literature, on a topic that opens up additional diagnostic possibilities in patients with left ventricular systolic dysfunction.
A particularly current topic is the setting of indications for coronary angiography.
It brings a new perspective on the application of various non-invasive readily available clinical variables such as ECG, echocardiography, and laboratory tests, especially if they allow a reasonable assessment of the likelihood of coronary heart disease in patients with ventricular dysfunction.
They are important because coronary angiography may be delayed in some patients.
The research is retrospective and all its significant shortcomings are highlighted.
The paper is written in correct English and although it may seem repetitive in some places, it is easy to read.

Author Response

Thank you very much for your comment. We deeply appreciate your positive feedback.

Reviewer 2 Report

This study attempted to address a diagnostic performance of coronary angiography to identify myocardial ischemia as the cause of left ventricular systolic dysfunction, namely ischemic cardiomyopathy, in a total of 536 patients without a medical history of coronary artery disease. Using a retrospective single center cohort study, they concluded a weak diagnostic performance of coronary angiography to detect ischemic cardiomyopathy except a subgroup showing high scores of their original grading system (i.e. SCS-HF) composed of young age, a presence of atrial fibrillation, female as negative factors, and hypercholesterolemia, diabetes mellitus, abnormal focal wall motion, q waves in electrocardiogram as positive factors. Regrettably, a quality of this study has the major limitation to consider a further evaluation for publication in this journal. First, a presence of coronary artery disease defined as 50% or over coronary stenosis did not have any clinical significance to have myocardial ischemia resulting in systolic dysfunction. Second, in the modern era of a great progression in a medical imaging modality, other non-invasive strategies using coronary computed tomographic angiography including an assessment of fractional flow reserve, cardiac magnetic resonance imaging including T1 mapping, and also cardiac scintigram including positron emission tomography are well-established to address the cause of cardiomyopathy. Finally, as widely published in ESC/ACC/AHA guidelines, an excellent risk estimation to predict atherosclerotic cardiovascular disease has already been established. 

Author Response

Thank you very much for your comments, your insights are very much appreciated. We totally agree that the presence of coronary artery stenosis does not necessarily imply causation, and therefore, it is not sufficient for the diagnostic of coronary cardiomyopathy. We think this is an important statement and that is why we point it out at line 172 and line 179 of the first manuscript submitted (182 and 189 of revised manuscript).

We also agree that nowadays there are non-invasive strategies that could help to address the cause of cardiomyopathy. However, coronary angiography remains the gold standard technic for detecting coronary stenosis, and for this reason it is used routinely by many cardiologists to rule out ischemic aetiology. On that line, our study could help to identify those patients with low risk of coronary stenosis in whom coronary angiography should be avoided and those patients with high risk in whom coronary angiography should be performed. We add this information in discussion section, line 170-174

As recently published 2021 ESC Guidelines of heart failure, coronary angiography may be considered in patients with left ventricular systolic disfunction with an intermediate to high pre-test probability of coronary artery disease and the presence of ischaemia in non-invasive stress tests for the diagnosis of coronary artery disease. That statement comes with a IIb class recommendation and level B of evidence. The only citation that comes with that recommendation is the STICH study (1) which was designed to evaluate the role of coronary artery bypass grafting in patients with heart failure and coronary disease. In that study, the performance of coronary angiography for detecting coronary stenosis in patients with heart failure was not evaluated and no score was presented to predict the presence of coronary stenosis. With all that, our study brings potential information that could help clinicians determine which patient might or might not benefit from an invasive approach. (We exposed the main characteristics of STICH study in the first manuscript submitted in line 209 (218 of revised manuscript) and now we added a few lines clarifying that this study was not designed to evaluate coronary angiography performance in line 233-236).

On the other hand, we think that use ESC or ACC guidelines for chronic coronary syndromes and their risk estimation criteria could led to an erroneous estimation since patients with heart failure are a different population and have their own risk and protective factors. By example, in our risk score the presence of atrial fibrillation/auricular flutter decreases the probability of coronary artery disease and that statement is not true (even contrary) for general population with suspected coronary disease.

  1. Velazquez EJ, Lee KL, Deja MA, Jain A, Sopko G, Marchenko A, Ali IS, Pohost G, Gradinac S, Abraham WT, Yii M, Prabhakaran D, Szwed H, Ferrazzi P, Petrie MC, O’Connor CM, Panchavinnin P, She L, Bonow RO, Rankin GR, Jones RH, Rouleau JL, STICH Investigators. Coronary-artery bypass surgery in patients with left ventricular dysfunction. N Engl J Med 2011; 364:1607-1616.

Of note we identify a mistake, and we made a minimum change in manuscript. In line 38 of the manuscript where it says “…level of evidence C.” we change it by “…level of evidence B”. In line 219 where it says “…(Level C) is an expert opinion…” we change it by “… level B…”.

Reviewer 3 Report

I would like to congratulate the authors on this work. And would like to ask few questions:

1) The authors state in the limitations that the sample size may not be adequate. How did they calculate the sample size in the first place? It was not mentioned in the methods section.

2) Table 1 in the supplementary material:

The total points of the derived score is 12: how was that calculated? If we calculate the individual points they will add to 20 (or 4 if we consider the negative marks)

3) The authors did not mention how we can use this derived score in practice. Would we consider those with intermediate and high scores as having ischemic cardiomyopathy? or just those with high scores?

4) How did the authors segregate the patients into low/intermediate/high?

5) It would be reasonable to determine the positive and negative predictive values of this score in detecting ischemic cardiomyopathy.

6) In the introduction (lines 44 and 45): the authors state "We hypothesize that coronary angiography in unselected patients with left ventricular dysfunction has a very relative utility for diagnosing ischemic cardiomyopathy". I don't understand what they mean by this statement. Do they mean it is not a standardized approach?

Author Response

 I would like to congratulate the authors on this work. And would like to ask few questions:

1) The authors state in the limitations that the sample size may not be adequate. How did they calculate the sample size in the first place? It was not mentioned in the methods section.

Thank you very much for your comment. We did an estimation of the principal known risk factors of coronary artery stenosis in the first 50 patients included. Then, we chose the risk factor with the lowest difference in prevalence between the two study groups. Thus, in the group with non-significant coronary stenosis we estimated a prevalence of hypertension of 68% and in the group with significant coronary stenosis a prevalence of 77%. The estimated sample size was 516 patients (alpha=5%; beta= 90%; two-sided). Therefore, our sample size is enough to support our conclusions. However, we are aware that this sample size it is not enough to perform an internal validation with sufficient confidence as we state in our limitations. 

We add a short version of this information in the manuscript; line 73-74.

2) Table 1 in the supplementary material: The total points of the derived score is 12: how was that calculated? If we calculate the individual points they will add to 20 (or 4 if we consider the negative marks)

Thank you so much for catching this error, which we have now corrected. We change total points for range of points.

3) The authors did not mention how we can use this derived score in practice. Would we consider those with intermediate and high scores as having ischemic cardiomyopathy? or just those with high scores?

Thank you for your comment. According to the results of our study, patients with a high score are at high risk of significant coronary stenosis and should be evaluated by coronary angiography and patients with a low score should avoid it due to low risk of coronary stenosis. However, patients with an intermediate score should be individualized and a non-invasive testing may be required before coronary angiography.

We add this information in discussion section, line 170-174

4) How did the authors segregate the patients into low/intermediate/high?

Thank you for your comment. We separated patients into three risk categories according to our SCS-HF risk score. The exact cut-off point for each category was determined by authors' criteria after careful analysis of our results. We supported our decisions by an analysis of ROC curve. In this analysis the best cut-off point for maximum efficiency was 7 points, which is why we chose it as the cut-off point for high-risk patients.

We add this information in methods section; line 94-96

5) It would be reasonable to determine the positive and negative predictive values of this score in detecting ischemic cardiomyopathy.

Thank you very much for your comment. Assuming a prevalence of significant coronary stenosis similar to our study (25%), a SCS-HF score of >6 points (the cut-off point of high-risk patients) represents a negative predictive value of 80.0% and a positive predictive value of 72.4%.

We add this information in results section; line 144-147

6) In the introduction (lines 44 and 45): the authors state "We hypothesize that coronary angiography in unselected patients with left ventricular dysfunction has a very relative utility for diagnosing ischemic cardiomyopathy". I don't understand what they mean by this statement. Do they mean it is not a standardized approach?

Thank you very much for your comment. It is true that this sentence may be difficult to understand and that’s why we change it for this sentence: We hypothesize that coronary angiography in unselected patients with left ventricular systolic dysfunction should be performed only in patients with higher risk factors and should be avoided in those without risk factors.

We add this information in introduction section; line 45

Round 2

Reviewer 2 Report

none